# Rational Design of Lipase ROL to Increase Its Thermostability for Production of Structured Tags

**DOI:** 10.3390/ijms23179515

**Published:** 2022-08-23

**Authors:** Jeng Yeong Chow, Giang Kien Truc Nguyen

**Affiliations:** Wilmar Innovation Centre, Wilmar International Limited, 28 Biopolis Road, Singapore 138568, Singapore

**Keywords:** *Rhizopus oryzae* lipase, thermostability, enzymatic interesterification, rational design, protein engineering, structured lipids, cocoa butter equivalent

## Abstract

1,3-regiospecific lipases are important enzymes that are heavily utilized in the food industries to produce structured triacylglycerols (TAGs). The *Rhizopus oryzae* lipase (ROL) has recently gained interest because this enzyme possesses high selectivity and catalytic efficiency. However, its low thermostability limits its use towards reactions that work at lower temperature. Most importantly, the enzyme cannot be used for the production of 1,3-dioleoyl-2-palmitoylglycerol (OPO) and 1,3-stearoyl-2-oleoyl-glycerol (SOS) due to the high melting points of the substrates used for the reaction. Despite various engineering efforts used to improve the thermostability of ROL, the enzyme is unable to function at temperatures above 60 °C. Here, we describe the rational design of ROL to identify variants that can retain their activity at temperatures higher than 60 °C. After two rounds of mutagenesis and screening, we were able to identify a mutant ROL_10x that can retain most of its activity at 70 °C. We further demonstrated that this mutant is useful for the synthesis of SOS while minimal product formation was observed with ROL_WT. Our engineered enzyme provides a promising solution for the industrial synthesis of structured lipids at high temperature.

## 1. Introduction

*Rhizomucor miehei* lipase (RML) and *Rhizopus oryzae* lipase (ROL) are 1,3-regioselective lipases that are widely used in the food industry to produce structured TAGs. These enzymes can target fatty acids on the sn-1 and sn-3 positions of a TAG molecule while leaving the fatty acid on the sn-2 position intact, allowing the production of TAGs with unique functions and nutritional values. An example of an important structured TAG is 1,3-dioleoyl-2-palmitoylglycerol (OPO), which is found in human milk and can be easily absorbed by infants to provide them with the essential nutrients for the development of cognitive and visual functions [1]. Production of OPO through the enzymatic inter-esterification (EIE) of plant oil has increased in recent years due to the increased demand for use of OPO in infant milk formulations. Another important use of 1,3-regioselective lipases is the production of 1,3-distearoyl-2-oleoyl-glycerol (SOS), which can be used as cocoa butter equivalent (CBE). The chemical composition of CBE resembles the composition of fats found in cocoa butter and hence CBE is widely used as a chocolate fat mimetic [2].

ROL has recently emerged as an important lipase for industrial applications due to its higher specific activity and 1,3-regioselectivity, in comparison with RML [3]. However, ROL has a lower thermostability and this limits the use of ROL to reactions that can occur at lower temperatures. To increase the thermostability of ROL (from *Rhizopus niveus*), Kohno et al., used error-prone PCR and a high-throughput halo assay to identify a ROL E190V mutant with a 15 °C increase in thermostability [4]. More recently, Zhao et al., used a combination of multiple sequence alignment and computation tools to design a ROL variant with an additional disulfide bond at the E190 and E238 positions [5]. This quadruple mutant (V209L/D262G/E190C/E238C) can retain about 50% of its activity at 55 °C for 1 day. Although these protein engineering efforts were able to increase the thermostability of ROL, most of these ROL mutants are still not suitable for EIE reactions that occur at even higher temperature. For example, production of OPO and SOS will require a minimum reaction temperature of 60 °C and 70 °C, respectively, due the high melting points of the palmitic acid and stearic acid used in the reaction.

Here, we describe the protein engineering of ROL to further increase its thermostability so that the enzyme can be used in EIE reactions at temperatures higher than 60 °C. To mimic the oil-rich conditions of an EIE reaction, we developed a spectrophotometric-based screening assay by using 2,3-dimercapto-1-propanol tributyrate (DMPTB) and 5,5-dithio-bis(2-dinitrobenzoic acid) (DTNB) [6]. Using information derived from the sequence alignment of ROL, we identified and constructed 26 single mutants of ROL and screened them for increased thermostability with the DMPTB/DTNB assay. Single mutants with higher thermostability were recombined by gene shuffling and undergo additional rounds of screening to identify mutants with even higher thermostability. After a few rounds of screening and selection, a variant of ROL containing 6 new mutations (F173Y/V175F/G177C/Q197H/F216Y/S267L) was identified that allows the enzyme to withstand temperatures up to 70 °C. We were also able to demonstrate that the enzyme can be used to catalyze the EIE reaction for the production of SOS at 70 °C.

## 2. Results and Discussion

### 2.1. Sequence Alignment of ROL and Visualization with WebLogo to Display Consensus Sequence

The sequence of our wild-type ROL (ROL_WT) shares a 97.8% sequence identity with the Uniprot entry P61871 (Appendix A). A sequence alignment of ROL_WT was created from the top 50 orthologous sequences of ROL (identified from BLAST) and visualized using WebLogo (Figure 1) [7]. The WebLogo of ROL provides us with useful information on the consensus sequence, degree of conservation and natural abundance of each residue. Other web-based servers such as HotSpot Wizard also rely on the use of amino acid conservation to help users identify hotspots that can be targeted for mutagenesis [8]. A total of 14 residues were identified for the first round of mutagenesis and screening (Table 1). These residues were selected based on their ability to form thermo-stabilizing interactions with a neighboring residue, or according to their natural abundance identified from the WebLogo. Since the ROL_4x mutant (E190C/V209L/E238C/D262G) has a higher thermostability than ROL_WT, the single mutants were created based on the ROL_4x template [5].

### 2.2. Expression and Screening of ROL_4x Mutants for Increased Thermostability

The pFAi2 vector used for the expression of the ROL mutants has a *BleoR* gene to confer resistance to Zeocin for plasmid selection in both *E. coli* and *P. pastoris* (Appendix A). It also possesses a GAP promoter and α-factor secretion tag for the constitutive expression and secretion of the lipase protein in *P. pastoris*, respectively [9]. This allows the rapid production and screening of the ROL lipases activity directly from the culture supernatant, without the need to undergo cell lysis and protein purification. The vector is also compatible to Golden-Gate cloning so that the ROL mutant libraries can be constructed and cloned into the vector easily [10,11].

Most of the screening methods described in the literature rely on the use of halo assays or PNP (4-nitrophenol)-esters for the rapid identification of lipase mutants with increased thermostability [4,12]. However, these methods often lead to many false-positive mutants as they involve the use of substrates that are fully solubilized in the aqueous phase. Lipases require an aqueous and oil interfacial layer for enzyme activation and hence will work best when the substrate is present in excess [13]. Hence, we developed the DMPTB/DTNB assay to facilitate the screening of the ROL mutants. Use of DMPTB has previously been described to measure the positional specificity of lipases [6]. Hydrolysis of DMPTB releases a free thiol group that can react with DTNB to produce an absorbance reading at 405 nm.

To compare the thermostability of each ROL mutant, we inoculated the cells harboring the mutants into BMDY medium for protein expression. The supernatants were recovered from the cell culture and subjected to heat inactivation for 1 h separately at 45 °C and 60 °C. After heat treatment, DTNB assay was used to measure the residual activity of each sample at the same temperature used for the inactivation. The residual activity measured at 60 °C was normalized against the activity measured at 45 °C for each mutant (Figure 2a). We selected 60 °C as the temperature for inactivation as ROL_4x is only able to retain 24% of its activity at this temperature (Appendix A). The top six mutants (N134Y, T136D, F173Y, Q197H, Q197Y and S267L) have a 2.2, 1.8, 1.8, 2.3, 1.9 and 2.4-fold improvement of activity when compared to ROL_4x, respectively (Appendix A).

Since the best single mutants only showed slight improvement in thermostability when compared to ROL_4x (less than 2.4-fold), gene shuffling was used to recombine the six mutations identified from the first round of screening. This allowed us to rapidly identify synergistic mutations that can lead to further improvement in the thermostability of ROL_4x. Golden-Gate cloning was used to assemble the combinatorial library and the library was transformed into *P. pastoris* for screening. After screening 500 clones with the DTNB assay, we identified ROL_7x (ROL_4x_F173Y/Q197H/S267L) with a much higher thermostability than ROL_4x and its single mutants (Figure 2b). ROL_7x can retain about 45% of its activity at 65 °C and has an 8-fold increase in thermostability over ROL_4x (Appendix A). It is interesting to note that the addition of N134Y and/or T136D mutations to ROL_7x does not lead to a further increase in its thermostability, even when the ROL_4x_N134Y mutant shows a 2.8-fold improvement in thermostability over ROL_4x (Appendix A). This suggests that the N134Y and T136D mutations are not able to act synergistically to improve the thermostability of ROL_7x.

### 2.3. Screening of ROL_7x Mutants for Increased Thermostability

Although ROL_7x has increased thermostability over ROL_4x, its activity drops significantly above 65 °C. Due to the high melting point of stearic acid, the enzyme can be used to catalyze SOS synthesis only if it can retain most of its activity at 70 °C. To further improve the thermostability of ROL_7x, we identified an additional 12 residues that can be targeted for mutagenesis (Table 1). These mutants were synthesized from the ROL_7x template and subjected to the same pipeline that was used during the first round of screening. To identify the improved mutants, heat inactivation was increased to 67 °C and 45 °C. At 67 °C, ROL_7x can only retain about 19% of its activity (Appendix A). From the first round of screening, we identified 3 single mutants with significant improvement in thermostability over ROL_7x (Figure 3a). The V175F, G177C and F216Y mutants have a 4.2, 6.4 and 2.7-fold increase in thermostability over ROL_7x, respectively (Appendix A).

Gene shuffling was then used to recombine these mutations for a second round of screening. After screening 100 mutants, we identified ROL_10x (ROL_7x_V175F/G177C/F216Y) with a higher thermostability over ROL_7x. This mutant has a much higher activity at 70 °C than 45 °C (121% increase in activity) and has a 54-fold improvement in thermostability over ROL_7x (Appendix A).

### 2.4. Large Scale Expression of ROL Mutants and Thermostability Assays

To confirm that the ROL mutants identified from the screening have increased thermostability and are able catalyze EIE at higher temperature, ROL_WT and the three mutants (ROL_4x, ROL_7x, ROL_10x) were subcloned and expressed in the pFAi2 vector containing an AOX1 promoter. This allows a higher yield and concentration of protein to be produced for the EIE reaction. The enzymes were expressed in *P. pastoris* using BMMY medium and the supernatants were concentrated 20-fold for enzymatic assays. SDS-PAGE analysis of the supernatants revealed that mature ROL is the dominant form of the protein produced (32 kDa) for ROL_WT, ROL_4x and ROL_7x (Appendix A). However, a 40 kDa ROL_10x can also be observed from the SDS-PAGE gel, which was identified to be the pro-form of ROL. Formation of pro-ROL is likely to be caused by the high expression level of ROL_10x, resulting in the inefficient processing and cleavage of the pro-ROL by the endogenous KEX2 protease [14]. Interestingly, some studies suggest that the pro-form of ROL has a higher hydrolytic activity and a difference in substrate specificity when compared to the mature form [15,16]. Quantitation of protein from the supernatant allows us to estimate the yield of ROL_WT, ROL_4x, ROL_7x and ROL_10x to be 47 mg, 40 mg, 40 mg and 120 mg per liter of BMMY medium, respectively.

The specific activity of each ROL mutant was measured by titration against the hydrolyzed products of HOSun (High Oleic Sunflower oil). At 40 °C, the specific activity of ROL_WT, ROL_4x, ROL_7x and ROL_10x were determined to be 900, 2900, 2900 and 4000 U/mg, respectively. Since ROL_WT is expected to be less thermostable than the other ROL variants, the lower specific activity of ROL_WT could be caused by the partial thermal inactivation of the enzyme during expression. To compare the thermostability of the mutants, the reaction was carried out at various temperatures ranging from 40 °C to 75 °C (in 5 °C intervals) and the specific activities of the enzymes were determined (Figure 4). As expected, ROL_10x was found to have the highest thermostability, followed by ROL_7x, ROL_4x and ROL_WT (Figure 4). This result is consistent with the results obtained from the DTNB assay, suggesting that the DTNB assay can be used as a quick and reliable method for the screening of thermostable lipases.

To confirm that the ROL mutants can catalyze the EIE reaction at higher temperature, tricaprylin (C_24_) and oleic acid (C_18_) were used as substrates for the reaction. Use of tricaprylin is advantageous since the compound does not give a strong background signal on the TLC plate after staining with iodine. Incorporation of one or two molecules of oleic acid into tricaprylin will results in the formation of the 1-oleoyl-2,3-dicapryl-glycerol (C_34_) or 1,3-dicapryl-2-oleoyl-glycerol (C_44_) products, respectively (Figure 5a). The products can be stained with iodine easily and are able to separate from each other on the TLC plate. This allows us to monitor the reaction and determine if the reaction has reached equilibrium. The reaction was carried with ROL_WT and its mutants with a temperature ranging from 40 °C to 80 °C. The products were then separated on the TLC plate for analysis (Figure 5b). The results from the TLC plate showed that ROL_WT can tolerate up to 45 °C, which is consistent with the hydrolysis assay carried out using HOSun. It was also observed that ROL_4x has the highest conversion efficiency at 50 °C for the EIE reaction, while hydrolysis can occur up to 60 °C (Figure 4). This suggests that the basis for increased thermostability is dependent on the type of reaction catalyzed by the enzyme. A possible explanation is that a two-phase organic-aqueous system is required for the EIE reaction, thereby increasing the rate of enzyme inactivation due to increased interfacial area of the reaction [17,18]. In addition, ROL_4x might require some form of thermal activation to catalyze EIE since less products were detected when the reaction was carried out at a lower temperature. Based on the Arrhenius concept, activation energy must be reached before product formation can occur for most enzymatic reactions. The observation suggests that ROL_4x has a higher activation energy than the other ROL mutants, which is a feature typically found in thermophilic enzymes. There are many examples of close enzyme homologues with very different activation energies [19]. Among the ROL mutants, ROL_10x was found to possess the highest thermostability and can catalyze the EIE reaction up to 70 °C.

To determine if ROL_10x can be used to produce SOS, EIE was carried out at 70 °C using HOSun and stearic acid as substrates. GC-FID analysis of the samples revealed that most of the TAGs present in HOSun are in the form of 1,2,3-trioleoyl-glycerol (OOO—67.1%) and 1-stearoyl-2,3-dioleoyl-glycerol (SOO—8.2%) (Figure 6). After incubation at 70 °C for 8 h, ROL_10x was able to produce 47.9% of SOS and 32.6% of SOO, while the relative percentage of OOO dropped to 8.6%. No conversion was detected for ROL_WT while ROL_4x only produced 2.8% of SOS and 15.5% of SOO. These results suggest that ROL_10x is suitable for use in the industrial production of SOS with reaction conditions reaching up to 70 °C.

### 2.5. Methanol Tolerance Assay and Production of Biodiesel

Since the thermostability of enzymes and their tolerance towards organic solvents are often well-correlated, we were keen to determine if the ROL mutants can tolerate high concentrations of methanol when compared to ROL_WT. Tolerance of lipases towards methanol means that they may have the potential to be used for the industrial production of biodiesel [20]. The commercial lipase Eversa (Novozyme) can tolerate up to 80% *v/v* methanol without any significant loss of activity [21,22]. Production of biodiesel was tested against ROL_WT, ROL_4x, ROL_7x, ROL_10x and Eversa using HOSun and methanol (50% *v*/*v*) as substrates. After separation of products on the TLC plate, we observed that only ROL_7x, ROL_10x and Eversa can produce significant amount of fatty acid methyl ester (FAME—biodiesel) (Figure 7a). To further evaluate the methanol tolerance of ROL_7x, ROL_10x and Eversa, various concentrations of methanol (50–90% *v*/*v*) were added to the reaction mixture. ROL_7x was found to retain most of its activity up to 50% *v/v* methanol while ROL_10x can retain its activity up to 60% *v*/*v* methanol. Although the methanol tolerance of both ROL mutants is lower than that of Eversa (80% *v*/*v*), we see a strong correlation between the thermostability and methanol tolerance of the ROL mutants. Such correlation has been observed in many other enzymes including lipases [23,24,25,26,27]. This finding suggests that further improvement to the thermostability of ROL might increase their tolerance towards organic solvent, which have the potential to be used for biodiesel production.

### 2.6. Mechanism of Increased Stability Observed in the ROL_10x Mutant

To provide a better understanding how the mutations in ROL_10x mediate its increase in thermostability and methanol tolerance, SWISS-MODEL was used to generate a homology model of ROL_10x based on the wild-type structure of ROL (PDB: 1LGY) [28,29]. It can be observed from the structure that most of the mutations found in ROL_10x are buried near the core of the enzyme, except for F216Y and S267L (Figure 8). Structural analysis revealed that the F216Y mutation forms an additional hydrogen bond with the carbonyl backbone of the N96 residue, thereby increasing the stability of the enzyme (Figure 8). On the other hand, S267 is located at the lipid binding region, which explains the high frequency of Leu at this position to aid the lipase in anchoring onto the lipid molecule and maintaining its overall structural stability (Figure 1).

Interestingly, the G177C mutant was found to be one of the best single mutants that can provide significant improvement to the thermostability of ROL (Figure 3). Structural analysis of ROL_10x revealed that C177 is unlikely to be involved in the formation of a new disulfide bond since no neighboring cysteine residue is present. However, cysteine residues are known to exhibit both polar and hydrophobic properties and can enhance the thermostability of enzymes based on hydrophobic interactions in addition to their ability to form disulfide bonds [30,31]. Hence, the improved thermostability of ROL_10x mediated by G177C is likely caused by an improvement to the hydrophobic packing of C177 and its neighboring residues, such as G148.

The V173F, F173Y and Q197H mutations are all located on the same side of two parallel β-sheets. Since there are at least 3 aromatic residues (F17, Y260, F261) within 5 Å of V175, the mutation of V175 to F175 could improve the hydrophobic packing within the region, through the formation of non-covalent pi-pi interactions (Figure 8). Similarly, the close proximity of Y173 and H197 could lead to the formation of a hydrogen bond, thereby strengthening the hydrophobic packing within the core of the enzyme [32]. The proximity of Y173 and H197 could also explain why both mutations can exert a synergistic effect to the increased thermostability of ROL_10x, while no improvement was observed for the N134Y and T136D mutants. The additional improvement contributed by these mutations could also be too low to be detected by the DTNB assay (Appendix A). A crystal structure of ROL_10x will provide us with a better understanding on the effects of these mutations and how they lead to the increased stability of the enzyme.

### 2.7. Concluding Remarks

Thermostability is one of the most sought-after properties of industrial enzymes as it can help to improve their shelf life and allows enzymatic reactions to be carried out at higher temperatures. Enzymatic reactions that are carried out at higher temperatures can increase the diffusion and reaction rates, and also help decrease the viscosity of the substrates [33]. In this study, we were able to increase the thermostability of the ROL lipase through multiple rounds of mutagenesis and screening. The new mutant (ROL_10x) has accumulated a total of six mutations and is now able to tolerate up to 70 °C. When EIE was carried out at 70 °C, using HOSun and stearic acid as substrates, a significant amount of SOS was produced. The enzyme can also retain most of its activity in the presence of 60% *v*/*v* methanol. These findings suggest that ROL_10x has the potential to be further developed for use in the food industries or the production of biodiesel [34].

## 3. Materials and Methods

### 3.1. Expression of ROL in P. pastoris

Visualization of the sequence alignment of ROL orthologues was carried out using WebLogo (http://weblogo.berkeley.edu/ accessed on 11 October 2021). ROL was codon-optimized and synthesized by Bio Basic in pFAi2-GAP vector for expression and screening (Appendix A). For small scale expression of ROL in *P. pastoris* using 96-deepwell plates, cells were first inoculated into 900 μL of BMDY medium containing 1% *w*/*v* yeast extract, 2% *w*/*v* peptone, 100 mM potassium phosphate, pH 6.0, 1.34% *w*/*v* yeast nitrogen base (Formedium), 2% *w*/*v* glucose, 0.00004% *w*/*v* biotin (Sigma) and 100 μg/mL Zeocin (InvivoGen). The cells were then incubated at 30 °C on a microplate shaker with shaking at 1500 rpm. After 3 days, the cells were pelleted by centrifugation at 4 °C (3000× *g* for 5 min) and the supernatant can be used directly for lipase assay.

To obtain ROL protein with higher yield and concentration, expression of the ROL gene is driven by AOX1 (methanol-inducible) promoter instead of the GAP (constitutive) promoter. Similar to the constitutive expression of ROL, cells were first inoculated into 30 mL of BMGY medium (replacing glucose in BMDY with 1% *v*/*v* glycerol) and grown overnight at 30 °C for the cells to accumulate a high cell density. The cells were then harvested and diluted to an absorbance of 6 (OD600) in BMMY medium (replacing glucose in BMDY with 1% *v*/*v* methanol). The cells were then incubated at 30 °C with shaking for an additional 3 days. An additional 1% *v*/*v* methanol was added to the cells twice on the first and second day of incubation. After 3 days, the supernatant can be harvested by centrifugation. The supernatant was then concentrated 20-fold to 1.5 mL using centrifugal filter units with a 10 kDa cutoff filter (Pall Corporation). Protein quantitation was carried out using Bradford reagent (Bio-Rad), by comparison with BSA standards. The size of the proteins was estimated using the ProtParam tool from Expasy (https://web.expasy.org/protparam/ accessed on 30 November 2021).

### 3.2. DTNB Lipase Assay for Screening

To screen for ROL mutants with higher thermostability, 50 μL of the supernatant containing ROL was first subjected to heat-inactivation for 1 h at various temperature. To set up the reaction, the following components were transferred to a 2 mL Eppendorf tube: 9.875 μL of C8/C10 methyl esters (methyl octanoate/methyl decanoate) (Wilmar, Indonesia), 0.125 μL of DMPTB (Sigma), 68 μL of Tris (100 mM, pH 8.0), 2 μL of DTNB (Sigma) (50 mM in DMSO) and 20 μL of supernatant. The reaction was incubated on a thermomixer at various temperatures with shaking at 2000 rpm for 1 h. The reaction was stopped by adding 125 μL of acetonitrile and 25 μL of Tris (2 M, pH 8.0). The absorbance of the reaction mixture was recorded using a microplate reader at 405 nm.

### 3.3. Gene Shuffling by PCR and Golden-Gate Cloning

The single mutants of ROL-4x identified from the first round of screening were combined by gene shuffling using Q5 DNA polymerase (NEB) for PCR. The following sets of templates and primers were used to generate three partial fragments of ROL that can be assembled by Golden Gate cloning using BsaI (NEB) and T4 DNA ligase (NEB). For the second round of screening, the following primers and templates were used to amplify two fragments of ROL from the single mutants of ROL-7x.

### 3.4. Determination of the Specific Activity of ROL by Titration

The specific activity of ROL and its mutants were determined by detecting the release of oleic acid from the hydrolysis of HOSun (Wilmar). Each reaction contained 150 μL of HOSun, 180 μL of 50 mM Tris, pH 8 and 20 μL of supernatant. After incubation on a thermomixer at 40 °C, 2000 rpm for 15 min, the reaction was stopped by adding 300 μL ethanol containing 0.1% *w*/*v* phenolphthalein. The mixture was then titrated against 50 mM NaOH until the purple color developed. To ensure initial rates were obtained, the supernatant was diluted so that the volume of NaOH added did not exceed 1 mL. The specific activity of ROL is calculated based on Equation (1).
(1)$$Specific\;activity\;\left(\frac{U}{mg}\right)=\frac{(V_{NaOH}\;Sample-V_{NaOH}\;Blank)\times \left[NaOH\right]}{T_{Rxn}\times E_{vol}}÷\left[E\right]$$
where *V_NaOH_ Sample* is the volume of NaOH added for the sample in mL, *V_NaOH_ Blank* is the volume of NaOH added for the blank in mL, [*NaOH*] is the concentration of the NaOH used for the titration in mM, *T_Rxn_* is the duration of the reaction in min, *E_vol_* is the volume of enzyme in mL and [*E*] is the concentration of the enzyme in mg/mL.

To determine the thermostability of the ROL mutants by titration, 50 μL of the supernatant was incubated on a thermocycler at selected temperature for 1 h. The reaction was then carried out at the same temperature used for the inactivation of the enzyme.

### 3.5. Analysis of EIE Reaction by TLC Assay

The EIE reaction was carried out by setting up the reaction in a 2 mL Eppendorf tube. The reaction contained 25 μL of Tricaprylin (Sigma), 50 μL oleic acid (Wilmar, Malaysia) and 25 μL supernatant (50 μg). The reaction was then incubated on a thermomixer set to 40 °C–80 °C, 2000 rpm for 1 h. The reaction was stopped by adding 750 μL isopropanol and 1 μL of the sample was spotted on the Silica gel 60 F254 TLC plate (Merck). The TLC plate was developed using a solvent system containing hexane and diethyl ether (80:20). After 10 min, the plate was air dried and stained with iodine.

### 3.6. Analysis of EIE Reaction by GC-FID

To detect the production of SOS by GC-FID, the EIE reaction was carried out in a 2 mL Eppendorf tube containing 50 mg HOSun (Wilmar), 100 μL stearic acid (Wilmar) and 50 μL enzyme (50 μg). The reaction was carried out at 70 °C, with shaking at 2000 rpm for 8 h. The reaction was stopped by heat inactivation at 95 °C for 10 min, followed by GC-FID analysis.

### 3.7. Methanol Tolerance of ROL Mutants

Production of biodiesel was carried out in a 2 mL Eppendorf tube containing 100 μL HOSun and 10 μL supernatant (concentrated). 10–90 μL methanol was also added to achieve a methanol concentration of 50–90%. The reaction was incubated on a thermomixer set to 40 °C, 2000 rpm for 24 h. The reaction was stopped by adding 900 μL isopropanol and 1 μL of the sample was spotted on the TLC plate. The TLC plate was developed using a solvent system containing hexane, diethyl ether and formic acid (80:20:1) for 5 min. The TLC plate was then air dried and stained with iodine for visualization.

## 4. Patents

A provisional patent for the use of ROL_10x was filed in Singapore (application number 10202201603Y) on the 18 February 2022.

## Figures and Tables

**Figure 1 ijms-23-09515-f001:**
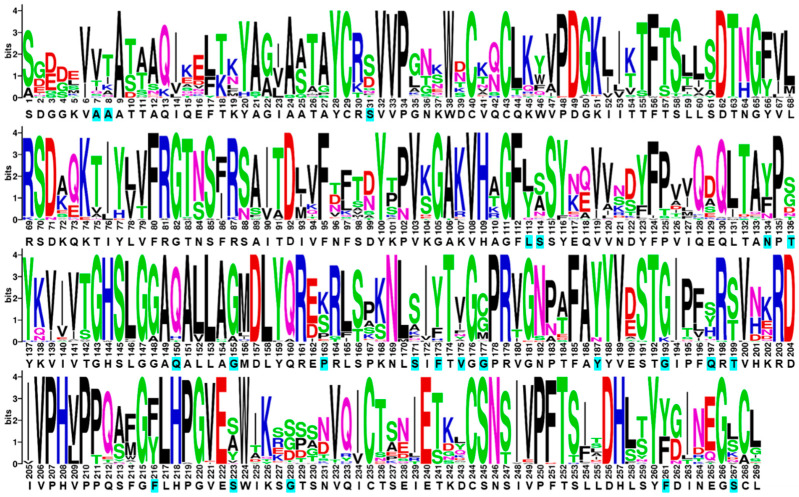
Weblogo of ROL. Positions of ROL targeted for mutagenesis are highlighted in cyan. Each logo consists of stacks of symbols, one stack for each position in the sequence. The overall height of the stack indicates the sequence conservation at that position, while the height of symbols within the stack indicates the relative frequency of each amino or nucleic acid at that position.

**Figure 2 ijms-23-09515-f002:**
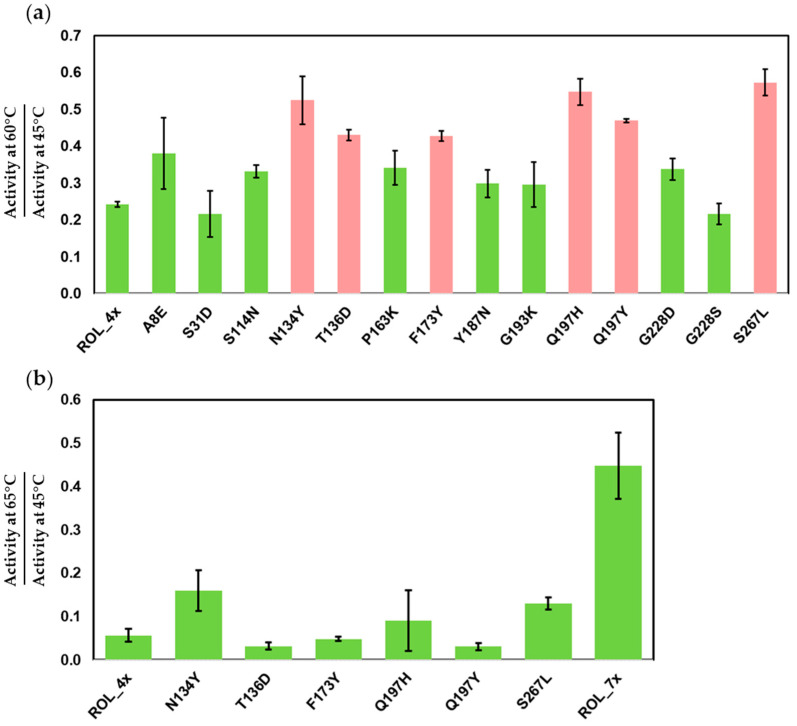
DTNB assay to identify the mutants of ROL_4x with increased thermostability. (**a**) Screening of single mutants of ROL_4x. Top mutants (red) are selected for gene shuffling. (**b**) Thermostability of ROL_7x compared to ROL_4x and its single mutants.

**Figure 3 ijms-23-09515-f003:**
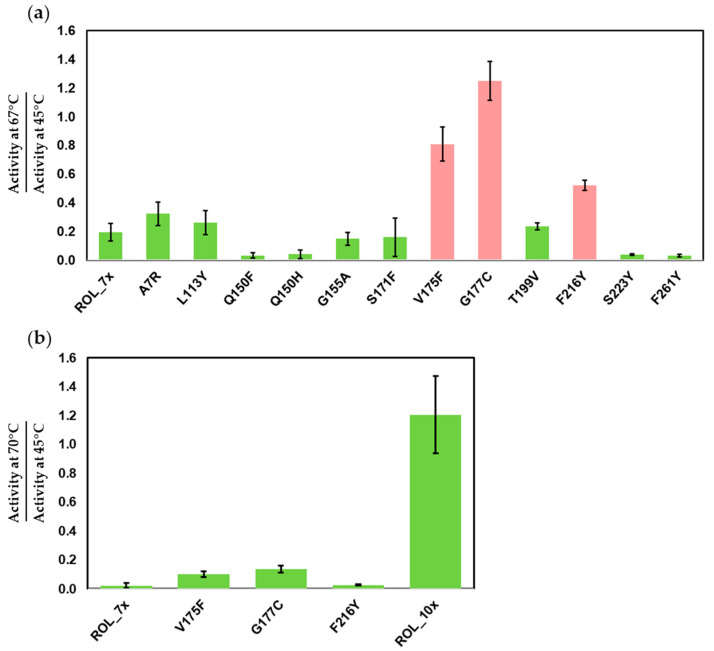
DTNB assay to identify the mutants of ROL_7x with increased thermostability. (**a**) Screening of single mutants of ROL_7x. Top mutants (red) are selected for gene shuffling. (**b**) Thermostability of ROL_10x compared to ROL_7x and its single mutants.

**Figure 4 ijms-23-09515-f004:**
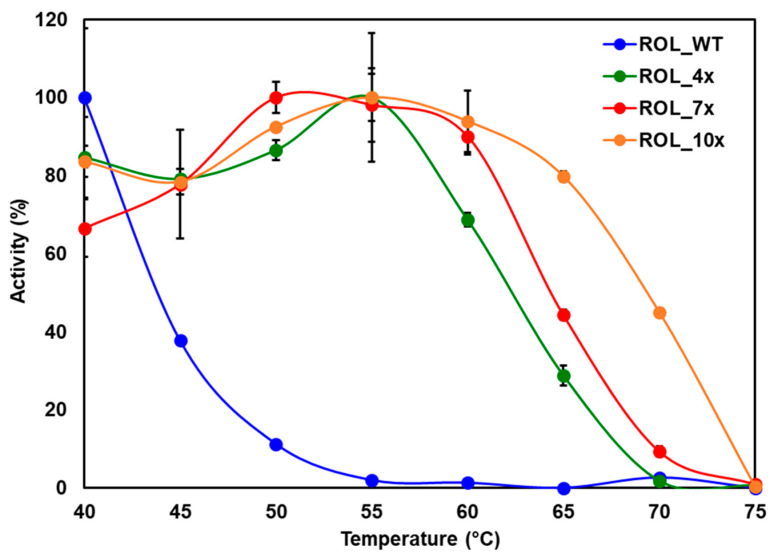
Thermostability of ROL mutants determined by titration against HOSun.

**Figure 5 ijms-23-09515-f005:**
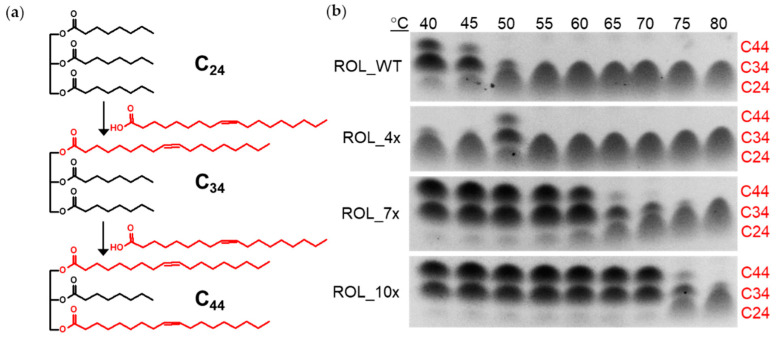
Use of TLC assay to monitor the EIE reaction. (**a**) Tricaprylin (C_24_) and oleic acid (C_18_) are used as substrates for the EIE reaction. The products from the reaction have a carbon number of C_34_ and C_44_, which can be separated and visualized on the TLC plate. (**b**) Thermostability of ROL mutants determined by TLC assay.

**Figure 6 ijms-23-09515-f006:**
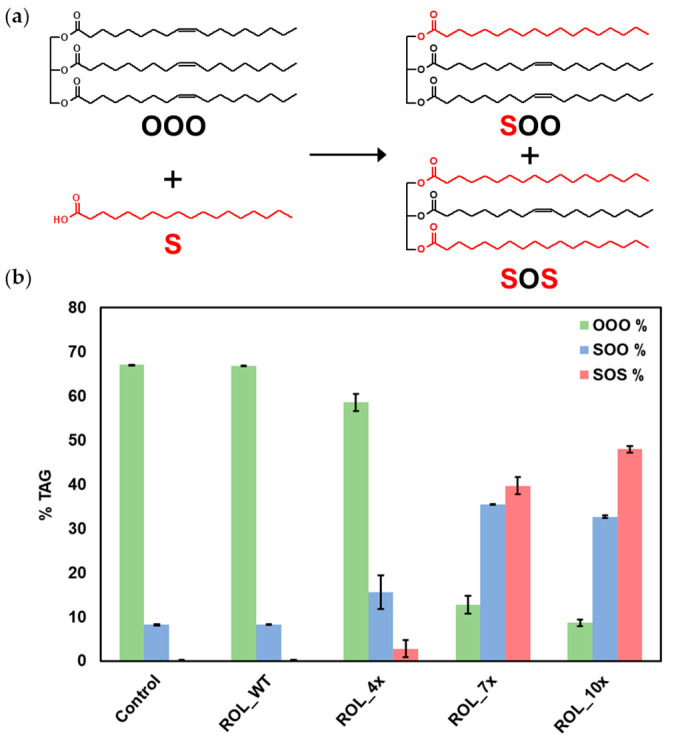
Production of SOS by EIE reaction. (**a**) HOSun (OOO) and stearic acid (S) are used as substrates for the reaction. (**b**) Relative percentage of OOO, SOO and SOS measured by GC-FID.

**Figure 7 ijms-23-09515-f007:**
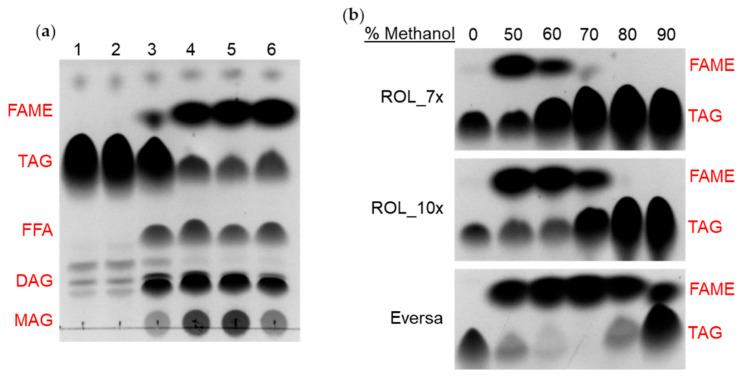
Production of FAME (biodiesel) and methanol tolerance of ROL mutants by TLC assay. (FAME) Fatty acid methyl ester; (TAG) Triacylglyceride; (FFA) Free fatty acid; (DAG) Diacylglyceride; (MAG) Monoacylglyceride. (**a**) Biodiesel production with HOSun and 50% methanol for (1) Control, (2) ROL_WT, (3) ROL_4x, (4) ROL_7x, (5) ROL_10x and (6) Eversa. (**b**) Methanol tolerance of ROL_7x, ROL_10x and Eversa.

**Figure 8 ijms-23-09515-f008:**
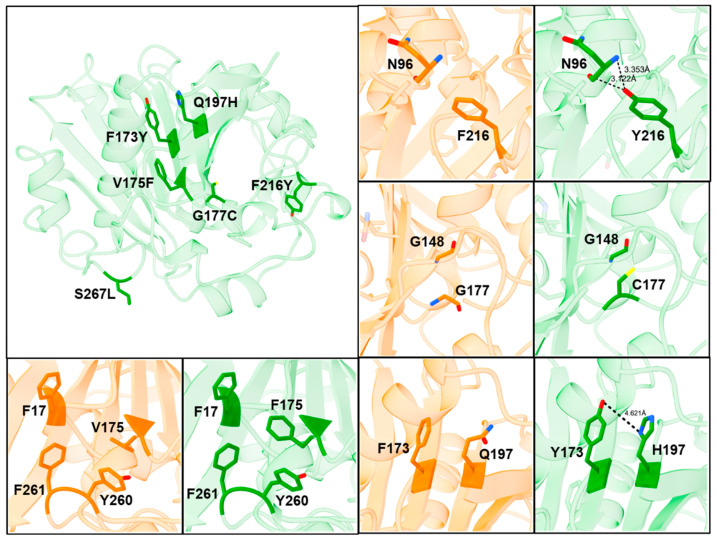
Mechanism of increased thermostability and methanol tolerance observed in ROL_10x. Structure of ROL_WT is colored in orange and structure of ROL_10x is colored in green.

**Table 1 ijms-23-09515-t001:** List of single mutants created for thermostability screening.

1 st Round Mutagenesis *		2 nd Round Mutagenesis **	
A8E	Y187N	A7R	V175F
S31D	G193K	L113Y	G177C
S114N	Q197H	Q150F	T199V
N134Y	Q197Y	Q150H	F216Y
T136D	G228D	G155A	S223Y
P163K	G228S	S171F	F261Y
F173Y	S267L		

* Mutants created from ROL_4x. ** Mutants created from ROL_7x.

## Data Availability

The data presented in this study are available in this article.

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
