# Peer review of "Rational Design of Lipase ROL to Increase Its Thermostability for Production of Structured Tags"

_ijms, 2022, doi:10.3390/ijms23179515_

Round 1

Reviewer 1 Report

I think that the paper under review deserves publication in the International Journal of Molecular Sciences. The topic is of interest for the journal readers and the described findings are scientifically sound. Overall, I deem that the paper is well written and I suggest correcting only few inaccuracies, in order to improve the quality of the work.

More specifically, I listed below some suggestions:

-Abstract, third line. ‘ .. has recently gained traction to be used …’ better  ‘…has recently gained interest because this enzyme possess high selectivity and …’

-Reference 1 should be placed next to 1,3-dioleyl-2-palmitoylglycerol (OPO). Thus four line before.

-The reference 4 do not describe a study on the increasing of ROL activity. The studied enzyme is a R. niveus lipase instead of a R. oryzae lipase. Please correct the text accordingly.

-The authors wrote ‘engineering efforts or engineering of ROL’. The description: ‘.. production of ROL through genetic engineering ..’ is more appropriate.

-Throughout the paper: the authors indicate the chains lengths writing C24, C34 , C44 and so on (eg. Figure 5). The carbon numbers should be written using subscript.

-References. The microorganism’s names should be written using italic. 

Author Response

Dear Reviewer,

            Thank you for the favorable review of our manuscript titled “Rational design of ROL lipase to increase its thermostability for production of structured TAGs”. I have revised the manuscript accordingly and have included a point-by-point response for your suggestions. Please see the file attached.

Yours sincerely,

Jeng Yeong

Senior Research Scientist

Wilmar International Limited

Reviewer 2 Report

I have reviewed the article Rational design of ROL lipase to increase its thermostability for production of structured TAGs and find it to be interesting work. In general, it is understandable and the methods and results are presented in an orderly manner. The introduction and conclusion allow us to understand the relevance of the work. However, for the article to be accepted for publication, it is essential that the discussion of results (the scientific support of the results) be reinforced. Everywhere in the results and discussion section are transcendental assertions, which are neither discussed nor supported by references. To mention something: lines 133-134; 199-200; 201-203 etc. All results should be reviewed and discussed in depth.

Author Response

(The authors gave the same response as above.)

Reviewer 3 Report

The manuscript describes the design and development of new mutants of ROL with increased thermostability and resistance to the methanol in the reaction medium that can be efficiently used in the synthesis of complex triglycerides (Such as SOS) and biodiesel. The work performed is highly interesting and suitable for publication on IJMS

Some minor change could be required prior publication. In fact, some part is not very clear and should be revised.

The approach used for screening of the thermostable mutants should be better explained because is not clear (line 112 and Fig 2).

After inactivation qt 45 and 60°C the residual activity was tested in two different experiments at 45 and 60°C?

Or the activity of the samples submitted to inactivation at 45 and 60°C were tested, after the inactivation time (1 hr) in the same condition?

Please better explain the sentence in line 199-200 for ROL4x. This enzyme can perform hydrolysis at temperature up to 60°C but the efficiency in the EIE reaction is occurring only at 50°C (fig5). At temperature lower and higher than 50°C the EIE reaction is not catalysed by this enzyme mutant, but this could be not related with its thermostability.

This enzyme behaves differently from the others that are able to catalyse the EIE reaction at all temperatures up to their stability temperature limit.

Probably for the ROL4x the mutation induces some change in the catalytic properties and not only in the thermostability of the enzyme.

Author Response

(The authors gave the same response as above.)

Reviewer 4 Report

This article outlines the design and mutation of lipase ROL to improve its thermostability, and the created variants ROL_10x could retain high activity under a high temperature. The sound results of the SOS biosynthesis were achieved from the engineered lipase, meaning the importance of this work.

Minor points & suggestions:

1. Line 297, what’s the temperature used for centrifugation?

2. Line 304, 6 OD? OD600=6? Also, Tris (2M, pH 8.0) is not one kind of buffer system (line 318), and it could be Tris-HCl (2 M, pH 8.0).

3. Is the gene of lipase ROL after codon optimization used for expression by P. pastoris? The specific activity of ROL and quantitation of protein from the supernatant are measured, and it is suggested to note the method of protein concentration.

4. In figures 5 and 6, it is recommended to show the chemical structure of molecules clearly.

5. Are the results of 32 kDa and 40 kDa from the software calculation? According to figure S4, pro-form of mutant ROL-10x is observed, can the protein cleavage occur when it lasts a long time?

6. HotSpot Wizard and WebLogo are used in section 2.1, the websites could be added to the materials section.

7. “ROL lipase” in the title may be “lipase ROL”, and “ROL_10x mutant” in the abstract may be “mutant ROL_10x”. Lines 163 and 164, the text “dominant form” should be “mature form”. Line 233, the text “the reaction” should be “the reaction mixture”, and the word “Aox1” (line 300) may be “AOX1”, etc. Also, several abbreviations may be clearly stated, such as PNP-esters (line 99), C34 and C44 (line 191), and C8/C10 methyl esters (line 314).

Author Response

(The authors gave the same response as above.)

Round 2

Reviewer 2 Report

The article can now be accepted for publication